# Investigating the Fibrillar Ultrastructure and Mechanics in Keloid Scars Using In Situ Synchrotron X-ray Nanomechanical Imaging

**DOI:** 10.3390/ma15051836

**Published:** 2022-03-01

**Authors:** Yuezhou Zhang, Dave Hollis, Rosie Ross, Tim Snow, Nick J. Terrill, Yongjie Lu, Wen Wang, John Connelly, Gianluca Tozzi, Himadri S. Gupta

**Affiliations:** 1Institute of Bioengineering and School of Engineering and Materials Science, Queen Mary University of London, London E1 4NS, UK; yuezhou.zhang@qmul.ac.uk (Y.Z.); wen.wang@qmul.ac.uk (W.W.); 2LaVision UK, 2 Minton Place, Victoria Road, Bicester OX26 6QB, UK; dhollis@lavision.com; 3Centre for Cell Biology and Cutaneous Research, Blizard Institute, Barts and the London School of Medicine and Dentistry, Queen Mary University of London, London E1 2AT, UK; rosie.ross@qmul.ac.uk (R.R.); j.connelly@qmul.ac.uk (J.C.); 4Diamond Light Source, Harwell Science and Innovation Campus, Didcot OX11 0DE, UK; tim.snow@diamond.ac.uk (T.S.); nick.terrill@diamond.ac.uk (N.J.T.); 5Centre for Biomarkers and Biotherapeutics, Barts Cancer Institute, Queen Mary University of London, John Vane Science Centre, Charterhouse Square, London EC1M 5PZ, UK; y.j.lu@qmul.ac.uk; 6School of Engineering, London South Bank University, London SE1 0AA, UK; tozzig@lsbu.ac.uk

**Keywords:** keloids, hypertrophic scarring, small-angle X-ray scattering, extracellular matrix, collagen fibrils, synchrotron X-ray imaging, digital image correlation, nanomechanics

## Abstract

Fibrotic scarring is prevalent in a range of collagenous tissue disorders. Understanding the role of matrix biophysics in contributing to fibrotic progression is important to develop therapies, as well as to elucidate biological mechanisms. Here, we demonstrate how microfocus small-angle X-ray scattering (SAXS), with in situ mechanics and correlative imaging, can provide quantitative and position-resolved information on the fibrotic matrix nanostructure and its mechanical properties. We use as an example the case of keloid scarring in skin. SAXS mapping reveals heterogeneous gradients in collagen fibrillar concentration, fibril pre-strain (variations in D-period) and a new interfibrillar component likely linked to proteoglycans, indicating evidence of a complex 3D structure at the nanoscale. Furthermore, we demonstrate a proof-of-principle for a diffraction-contrast correlative imaging technique, incorporating, for the first time, DIC and SAXS, and providing an initial estimate for measuring spatially resolved fibrillar-level strain and reorientation in such heterogeneous tissues. By application of the method, we quantify (at the microscale) fibrillar reorientations, increases in fibrillar D-period variance, and increases in mean D-period under macroscopic tissue strains of ~20%. Our results open the opportunity of using synchrotron X-ray nanomechanical imaging as a quantitative tool to probe structure–function relations in keloid and other fibrotic disorders in situ.

## 1. Introduction

Fibrotic scarring characterizes a range of biological disorders, including: cutaneous scarring; keloid formation; fibrotic capsules in cancer; burns; and cardiovascular injuries and involved substantial pathological remodeling of the collagen-rich extracellular matrix (ECM), and significant patient health outcomes [1,2]. It is believed that altered fibroblast phenotype, modified matrix-level mechanics in the ECM and genetic factors act in concert to control fibrotic progression [3,4]. The in vivo tissue biomechanics are an important factor mediating fibrosis development, scar formation, or resolution [5,6]. However, limited experimental data exist on small-scale (collagen fibril level) mechanics in fibrosis. Lower maximum force has been reported [7], but deformation mechanisms of the collagenous fibrillar ECM have not been explored. Developing probes of the fibrillar ECM mechanics in fibrotic conditions would enable early onset detection and understanding mechanobiological pathways. In this paper, we demonstrate a nanoscale X-ray mechano-imaging method to measure spatially resolved fibrillar-strains in fibrotic tissue using keloid scarring as an example.

Keloids are an example of a fibrotic skin tissue disorder related to hypertrophic scarring [8,9], whose etiology can involve genetic factors, cytokines, transcription factors and growth factor signaling [10,11,12,13]. Keloid ultrastructure shows the presence of an abnormal cartilage-like composition, including collagen type II and aggrecan, and over 20 keloid-unique proteins [14]. The biomechanics of this altered ECM at the fibrillar level will affect keloid development, as keloids often form in areas of tight skin or high mechanical stress [15].

To determine fibril-level biomechanics (strain, reorientation, and intramolecular ordering) in the ECM of such fibrotic tissues, small-angle X-ray scattering (SAXS), particularly coupled with a synchrotron source, is a powerful tool [16,17,18,19,20,21,22,23]. In fibrillar collagen, SAXS measures the axial periodicity (D~65–67 nm) in electron density along the fibril axis, to obtain a series of meridional Bragg peaks in the wavevector range of 0.1–1 nm^−1^. Changes in the axial peak position are linked to fibril strain [18,24], while changes in the angular intensity patterns are linked to fibrillar reorientation [23]. Additionally, changes in peak intensities are linked to a combination of collagen content, the arrangement of molecules inside the fibril, and 3D orientation [16,25,26]. In fibrotic scarring from dehydrated sections of burn tissue, static (undeformed) SAXS patterns have recently been acquired showing differences in fibrillar alignment across the wound [27,28]. However, SAXS has not been used to explore the in situ fibrillar-level mechanics in fibrosis.

The application of SAXS to the biomechanics of spatially and compositionally heterogeneous fibrotic tissues such as keloids brings several challenges. First, due to spatial–structural heterogeneity, a localized SAXS pattern (with synchrotron a spatial resolution of ~10 μm) will not be representative of the tissue structure, necessitating SAXS-mapping experiments with a micron-sized beam [29]. Second, the material-level tissue heterogeneity will lead to heterogeneous deformation at the microscale; for example, regions with higher density of stiff collagen fibrils, or more aligned fibrils, will be less strained. This poses a basic technical problem for microfocus scanning SAXS probes of fibril mechanics: to determine the strain, the same tissue volume must be compared before and after deformation. If the tissue deformation is microscopically inhomogeneous, identifying the SAXS scan points corresponding to the same tissue volume is not possible by using macroscopic strain measures to predict (as shown in bone, e.g., [30]). Finally, the altered ECM composition and increased proteoglycan content relative to normal skin [14] may introduce additional nanoscale phases with their own scattering contrast.

Here, we demonstrate how to overcome this challenge by showing a proof-of-concept combination of microfocus scanning SAXS mapping, multi-component modelling, in situ mechanical testing, and a variant of digital image correlation (DIC) using a diffraction-contrast texture. Applications of DIC [31] and its three-dimensional counterpart (digital volume correlation or DVC) on biological tissues have grown rapidly in recent years [32,33,34]). In their standard form, this class of methods measures deformation at the microscale using correlation displacements of surface or volumetric texture measured with digital cameras or tomographic imaging [33]. For hydrated tissue samples, the application of external surface texture or speckle can be difficult. Our idea is to use the intrinsic SAXS imaging contrast (acquired during 2D SAXS mapping) to circumvent this problem. By generating 2D SAXS maps before and after deformation, and using these as input to DIC software, we will track the displacement of tissue locations by correlating points with the same nanoscale SAXS scattering contrast. Concurrently, identifying the separate SAXS peaks arising from both collagen D-period and other periodic spacings will allow image-based SAXS contrast mapping for different phases. The irregular and disordered tissue structure of keloids at the micron scale [35] makes them well-suited as a tissue type on which to develop the method. The goal here is to demonstrate how nanoscale deformation in the ECM components of heterogeneous connective tissues can be mapped with microscale resolution.

## 2. Materials and Methods

### 2.1. Materials

Keloid samples were obtained from four patients from the plastic surgery department at Barts Health NHS Trust. All subjects gave informed consent, and the study was conducted under local ethical committee approval (East London Research Ethics Committee, study no 2011-000626-29). Multiple keloid sections from the four patients were measured; to demonstrate the technique, results from two samples from two patients are reported here—the first set of experiments will show the nanoscale contrast from different phases, and the second the strain-measurement in inhomogeneous deformation.

### 2.2. Sample Preparation

Using twin surgical blades in a customized holder with a 1 mm thickness, keloid sections were prepared from the intact 3D keloids, along the vertical direction to the skin surface. The approximate dimensions were 30–35 mm in length, 10–15 mm in width and 1 mm in thickness [25]. Keloid samples that were not tested immediately were wrapped in gauze soaked with phosphate-buffered saline (PBS), placed in petri-dishes, and stored under −20 °C in the frozen state. A subset of samples was scanned using microfocus SAXS without loading, and another subset was scanned in combination with in situ tensile testing.

### 2.3. SAXS Data Collection

A Pilatus 2M detector with a pixel size of 172 μm and a resolution of 1475 × 1679 pixels was used to collect SAXD data at beamline I22, Diamond Light Source, Harwell Science Campus, Didcot, UK [36]. Before SAXD scanning, the sample-to-detector distance and beam center were elucidated using a silver behenate calibrant; the sample-to-detector distance was calculated to be 5.5 m. Each SAXS measurement was for 1 s, with an X-ray energy of 14 keV and a beam size of ~15 μm.

### 2.4. In Situ Tensile Testing with SAXD

Uniaxial tensile tests, in a strain-controlled mode, were carried out on a customized mechanical tester (Figure 1a). The nominal (grip to grip) strain was measured by a DC-encoder motor (M126 DG; Physic Instruments, Teddington, UK) with a 110 N tensile loading cell (RDP Electronics Ltd., Wolverhampton, UK) controlled by a LabVIEW program (National Instruments, Newbury, UK). Note that it was possible that the actual tissue strain would differ from the grip-to-grip strain due to potential sliding between grip and sample, and it was therefore measured using a special image correlation method, which is described in the next section. For hydration, PBS was placed on the sample, and Kapton film sections (DuPont Ltd., Wilmington, DE, USA) were placed on both sides of the sample. Due to surface tension, the films formed a stable leakproof construct around the tissue but did not artefactually affect the load readings as they glided on the fluid surface [37]. After samples were fixed on the sample holders, a 0.1 N tare load was applied to make sure the samples were taut. The samples were loaded to failure with a constant velocity of 0.005 mm/s and entire tensile loading procedure of one sample took ~6–7 min. The tensile data of load–displacement curves were recorded while at specific strain levels—0%, 20% and 40%. The stretching process was paused during SAXD scanning. As above, to demonstrate the method, only the comparison between the 0% and 20% tissue strain is shown in the current report.

### 2.5. Small-Angle X-ray Diffraction Data Analysis

A 2D SAXS pattern of the scanning image was collected and integrated using DAWN [38] to obtain the 1D scattering intensity profiles versus the wavevector q (radial plots, I(q)) or azimuthal angle χ (azimuthal plots, I(χ)) following our previous protocols (17).

Collagen SAXS analysis: The 5th order meridional collagen peak (q = 5 × 2π/D) was used for analysis to find the D-period and peak intensity. The 5th order peak was used for analysis, as the diffuse SAXS background is near-linear around the 5th order wavevector range (0.40–0.52 nm^−1^), in contrast to a more steeply sloping, nonlinear behavior at the lower wavevectors (~0.25–0.35 nm^−1^) characteristic of the 3rd order peak (the other meridional peak with high intensity). A Gaussian-peak model combined with an exponential decay model of diffuse background intensity was applied for peak fitting of the 5th collagen order in I(q) to calculate the D-period and axial peak width w_q_ (Figure 1d). A three-Gaussian-peak model was applied for fitting the two 180°-separated peaks in I(χ) arising from the 5th meridional collagen order to determine mean fibril angle χ_0_ and azimuthal peak width w_χ_. The reason for the three peaks was to correct for wrap-around effects arising from the 0°→360° equivalence. For both the radial and azimuthal plots, the 1D profiles were batch-fitted by customized Python scripts, using the non-linear, least-squares, fitting package *lmfit* [39] with user-defined model functions.

Interfibrillar component (IFC) SAXS analysis: The 2nd order broad IFC peak(q = 2 × 2π/D_IFC_) was observed and used for the analysis to find the spatial period D_IFC_ and integrated peak intensity. A Gaussian-peak model combined with an exponential decay model of diffuse background intensity was applied for peak fitting of the 2nd IFC order in I(q) to calculate the D-period. Again, these I(q) profiles were batch-fit by customized Python scripts, using *lmfit* with user-defined model functions.

Total, collagen peak and IFC peak intensity: The total SAXS intensity I[total] is the area under the I(q) curve and is related to the amount of interfacial area between the nanoscale constituents (fibrils) and extrafibrillar matrix. The collagen peak intensity I[c] is the peak area (after diffuse SAXS background subtraction) of the 5th order meridional peak. Similarly, the IFC peak intensity I[ifc] is the peak area (after diffuse SAXS background subtraction) of the 2nd order IFC peak.

### 2.6. Digital Image Correlation (DIC) Analysis Using SAXS Diffraction Contrast

The collagen peak intensity maps were rendered as images and processed via the Digital Image Correlation technique using DaVis 10.1.2 (LaVision GmbH). The processing scheme employed 25 × 25 pixel subsets, with a 3-pixel step size. This is an unusually small step size, but was necessary to obtain a reasonable number of displacements across the low-resolution (by the standards of DIC) collagen peak SAXS mapping image. The subsets used round weighted windows and a 6th order spline for gray scale interpolation. The strain window used to calculate the local strain used 7 × 7 subsets (i.e., a fit was performed on the displacements (u_x_, v_y_) calculated over the local 7 × 7 subsets). It was not possible to ascertain levels of displacement uncertainty using the accepted approach of a zero-displacement test, as there was no repeated acquisition of images with zero movement (to avoid overexposure of the sample to the X-ray beam).

## 3. Results

Using scanning microfocus SAXD and micromechanical tensile testing (Figure 1a), SAXD diffraction data were collected and used to determine the distribution and mechanics of fibrillar-level components in keloid tissue. A typical 2D SAXD pattern on keloid is shown in Figure 1b, with the radial and azimuthal integrations in Figure 1d,e. For the collagen D-period, we can estimate the collagen peak intensity, orientation of fibrillar components, and pre-strain level of collagen as shown in Figure 1c–f. The fifth order collagen peak was selected for radial and azimuthal integration, following our past practice for cartilage [16]. A Gaussian with linear sloping background was used to fit the D-period peak (Figure 1d, inset). For the example shown, q_05_ = 0.4812 nm^−1^, leading to D = 10π/q_05_ = 65.29 nm. For the collagen fibril orientation, the *I*(χ) peaks could be fitted similarly, using two Gaussians separated by 180°, which for the example shown, has the peaks at ~161° and 341°.

While the sharp meridional arcs, in a near-horizontal direction in the example shown, correspond to the fibril D-period (D~64–67 nm for collagenous tissues [21]) and are expected for a collagenous tissue, a new feature specific to keloids is a diffuse set of reflections orthogonal to the main direction of the meridional arcs. A 0–360° azimuthal integration (Figure 1c,d) shows these two sets of reflections more clearly. The diffuse orthogonal peak is seen to have a fundamental repeat in the range of 88–100 nm, with the lowest order (*n* = 1) partly masked due to the scattering of the direct beam from the beam-stop (Appendix B Figure A1). Since fibril radii in collagenous tissues are typically between 50 to 200 nm, it is plausible that the peak represents an interfibrillar spacing (orthogonal to the fibril axis) due to strong scattering contrast between the collagen fibril and a keloid-specific interfibrillar phase of indeterminate composition, which may consist of proteoglycans such as aggrecan, as shown recently [14]. As shown in Figure 2 and later, the intensity of the interfibrillar component peak (hereafter abbreviated IFC) is highly variable across the tissue, indicating that the IFC-phase varies in density across the tissue.

By collecting intensity values from SAXD patterns in each position, we have made mapping images of the SAXS peak intensity from collagen *I*[c] and the interfibrillar component IFC *I*[ifc], which can be compared to total SAXS intensity *I*[total], shown in Figure 2 for an example keloid section with the epidermis toward the top of the Figure. Considerable microspatial heterogeneity of the nanofibrillar components is shown, with *I*[total] (Figure 2a) at relatively high intensity bands in the upper third of the keloid tissue close to skin surface, while the deep part showed low intensity. In contrast to the somewhat smooth *I*[total] pattern in Figure 2a, *I*[c] and *I*[ifc] (Figure 2b,c) show a grainier, streak-like structure which, in the case of *I*[c], is dominated by a diagonal band from the top right to lower left (Figure 2b), and for *I*[ifc] by a high-intensity horizontal band at the skin surface, coupled with high-intensity narrower spots along the right-hand side.

Beyond the peak intensity for collagen and IFC, the peak positions provide measures of axial meridional spacing (D-period; for *I*[c]) and interfibrillar spacing (D_IFC_, for IFC). The D-period and D_IFC_ were calculated at each scan point and rendered as 2D color mapping images (Figure 3). A threshold-based representation, showing calculated values only when *I*[c] and *I*[ifc] exceed minimum scattering thresholds, was used to avoid artefactual D-spacings from points with little or no peak intensity. In the collagen D-period mapping, the D-period varied from ~64.94 nm to ~65.10 nm (a 0.23% variation). The physical meaning of a lower D-period is that collagen fibrils have reduced tensile pre-strain (i.e., relatively relaxed or flexible), or have axially tilted intrafibrillar molecular arrangements [40]. It is worth noting that collagen fibril D-period changes under loading are often small, e.g., up to 0.3–0.5% in bone [37]; therefore, even apparently small changes such as the 0.23% variation are significant, corresponding to ~1 MPa using 500 MPa as the collagen fibril modulus [41]. In the IFC mapping of D_IFC_, variation in DIFC was considerably more, ranging from 88 nm to 100 nm (~16% variation). Similar to the D-period representation, the top regions with high *I*[ifc] also showed low pre-strain (D_ifc_), while in middle section, pre-strain was relatively high. To summarize, the regions with rich fibrillar components have lower initial pre-strain tendency; such local tissue regions could be extended more during the loading process.

To explore the spatially resolved fibrillar response to tensile load in keloids, we combined micromechanical testing with in situ SAXS (Figure 4a,b). We note that for the example shown, the collagen scattering was high throughout the sample (in contrast to Figure 2 and Figure 3), and a dense (rather than sparse, as in Figure 3) grid of collagen peak intensities could be detected. Tensile strain was applied from a 0% strain to 20% strain level, with 2D SAXD scans at the two strain levels (for 20% strain, after 5 min of stress relaxation). During the stretch phase, the tensile deformation was applied via the top sample holder, and the 2D SAXS scans at a 0% and 20% strain level were therefore aligned from the bottom (reference edge) to show the shift. Comparing the 0% and 20% strain maps in Figure 4a,b, it is evident that local increases of collagen peak intensity *I*[c] occur upon stretching. Stress-induced in-plane realignment is evident from the two inset SAXS scans. Before stretching, in-plane orientation directions of collagen fibrils are less aligned with respect to the vertical direction, while after stretching, the collagen fibrils are more vertically oriented, and the range of angles also narrows. The microspatial heterogeneity in fibrillar components is evident in regions with low *I*[c] (voids), which may be enclosing sub-regions rich in another matrix component e.g., lipids, elastin, or other ECM components. Figure 4c,d show the statistical distribution of collagen D-period and orientation angles before (green) and after (orange) loading, plotted via kernel density estimators and histograms. From Figure 4c, the D-period mode for collagen fibrils at both a 0% and 20% strain was 64.7 nm, but an increase in mean D-period from 64.7 nm (0%) to 64.8 nm (20%) is observed. The shape of the kernel density estimator curve changes slightly, as expected from the inhomogeneous tissue structure. For orientation directions (Figure 4d) a narrowing of the distribution around the vertical 90° direction is seen.

Since the deformation and strain levels change across the tissue, fibrillar-level comparisons require the correlation of undeformed and deformed SAXS-scattering grid points. Using digital image correlation (DIC) on the collagen SAXS peak intensity as a contrast measure (Figure 1f), the displacement (u_x_, v_y_) for each point on the undeformed grid was calculated (as described in Materials and Methods). Using these displacement fields, Figure 5a,b shows the vertical displacement of a horizontal line of SAXS scan points (red rectangles, left) to an irregular, downward sloping line after deformation (red rectangles and dashed line shape, right). To demonstrate the heterogeneous nanoscale deformation behavior, four selected SAXS points (*, **, § and §§) were compared before and after deformation in Figure 5c,d. A spectrum of patterns is seen in the radial intensity profiles *I*(*q*); these include (a) an increase in collagen peak intensity but no peak shift (left: *), (b) an increased D-period as well as peak broadening (increased variability in D; **), (c) minimal changes, and (d) a decrease in collagen peak intensity. Correspondingly, the angular intensity plots show variable behavior as well, with (a) *: a decrease in peak width indicating increasing concentration of fibrils toward the loading axis, (b) **: an increase in peak width (indicating broadening), (c) §: minimal change, and (d) §§: a shift of the main fibril orientation direction towards the loading axis. Line profiles of these nanoscale ECM parameters, in the deformed and undeformed states, are shown in Figure 6.

## 4. Discussion

The main results of the current work are:The identification of a spatially heterogeneous, two-phase nanoscale mixture in keloid tissue seen in SAXS, with the narrow meridional peaks from collagen at D~65 nm combined, with broader peaks arising from an interfibrillar IFC component with a spacing of ~90–100 nm (Figure 2 and Figure 3);An increase in fibril D-period and the reorientation of fibrils upon application of external strains, characterized by a change in distribution of the D-period (Figure 4).The demonstration that digital image correlation using SAXS collagen contrast can be used to track spatially inhomogeneous tissue regions at the microscale (Figure 5) which can allow for the measurement of nanomechanical parameters such as D-period changes of the same tissue volume before and after loading (Figure 6).

In more detail, our results revealed that at the nanoscale, the X-ray scattering contrast from the keloid ECM comes from a two-phase mixture of (1) oriented collagen fibrils interspersed with (2) a second, interfibrillar phase (denoted by IFC) with strong scattering contrast in electron density relative to the fibrils. Further, the keloid scar exhibits a high degree of spatial heterogeneity in the nanoscale parameters, likely arising from the spatiotemporal tissue formation dynamics in keloids, as noted in prior work [42].

Quantitative parameters of the meridional diffraction pattern that can be compared from collagen in keloids to prior results from SAXS on collagen in cartilage [16], skin [25] and tendon [43,44], are average D-period and the ratio of peak order intensities. As shown in Table 1, the average D-period in keloid is close to static (unstretched) skin, somewhat lower than Type II collagen in deep-zone cartilage, Type I collagen in bone and in wet rat tail tendon (D~67 nm). While hydration changes the D-period [45] (dry rat tail tendon has D~65 nm versus the wet value of D~67 nm), this factor does not affect our results, as samples were kept hydrated in a transparent foil during the experiment. The variation in D-period is likely linked to intrafibrillar conformation of the tropocollagen molecules and tilting of the microfibrils [46]. In tissues such as skin and cornea, the microfibrils are tilted in respect to the main fiber axis by angles of up to 15–20° [47]. Atomic force microscopy studies have suggested a coiled rope-like structure for fibrils [48], and tissues where the unstressed D-period (keloid tissue ~65 nm) is less than that of wet tendon (~67 nm) will likely have the microfibrils tilted away from the fibril long axis. Such fibril tilts may also be significant in interpreting the nanoscale mechanical changes, as discussed below.

The collagen peak intensities of the meridional SAXS pattern can be linked to the gap/overlap ratio of tropocollagen molecular packing and intrafibrillar disorder [16,21]. It is known from previous work that peak intensity patterns are sensitive to hydration, with odd orders more intense in hydrated collagen and even orders stronger in dry collagen (as seen in tendon [49]). By comparing intensities with respect to a reference peak (fifth order; Table 1) for keloid, tendon (wet and dry), dehydrated scar and wound tissue, skin, and bone, the results show differences in numerical values for keloids compared to other tissues. I_3_/I_5_ is larger in keloids than for other hydrated tissues, with odd order peaks dominant over even orders (which is characteristic of hydrated collagen). The difference between wet and dry collagen can be seen by comparing I_4_/I_5_ for the dry- and wet tendon data in Table 1, which also presents recent results on dehydrated scar tissue [28], showing similarly high even/odd ratio I_6_/I_5_.

For the second set of broad equatorial SAXS peaks orthogonal to the meridional pattern, these are characteristic of interfibrillar periodicity orthogonal to the fibril axis (Figure 1). The equatorial periodicity length is similar to the fibril diameter found in collagenous tissues (ranging between 50 to 200 nm) [50], with a mean value of ~90–100 nm (Figure 3b). The characteristic length-scale of this phase should be slightly larger than the fibril diameter, if the fibrils are closely packed. Recent electron microscopy analysis of collagen fibrils by Zhou et al. [7] have reported a mean diameter value of ~70 nm for keloid tissue versus ~120 nm for extra-lesional skin, which is, as expected, slightly smaller than the expected IFC spacings found here, and indicates an interfibrillar spacing of ~20–30 nm in regions with a significant amount of the IFC scattering component. The electron density difference between the interfibrillar matrix (PGs, elastin, NCPs, water) and the collagen fibrils contributes to the strong X-ray contrast leading to clear IFC peaks. The wider IFC peak width relative to the meridional (IFC: 0.016 vs. collagen: 0.003 nm^−1^) may be due to heterogeneity in fibrillar radii (also seen in [7]).

The heterogeneous nanostructure of keloid tissue is shown in Figure 2 and Figure 4, with intra-keloid variation in the integrated collagen meridional peak intensity *I*[c] and the IFC peak *I*[ifc]. The variation arises from factors including different concentration levels of collagen and IFC phase, orientation of the fibrils within the scattering volume, and intrafibrillar disorder and the gap/overlap ratio. Increasing the collagen fibril content will increase I[c]. However, anisotropic 3D scattering of collagen fibrils [51] means peak intensity depends on the Ewald sphere intersection with this anisotropic reciprocal-space scattering (the measured 2D pattern). For fibrils tilted at large angles (>30–40°) relative to the X-ray beam (perpendicular to the tissue surface), the meridional scattering will barely intersect the plane with a low detector intensity even if the density of collagen fibrils is high. A second, intrafibrillar effect on the SAXS pattern arises from axial disorder and the gap/overlap ratio in the fibrils [16,21]. If intrafibrillar molecular stacking is disordered, leading to a fuzzy gap/overlap interface, peak intensities will decrease [16,21]. Further, odd- and even-order Bragg peaks will change in opposite directions if the length of the gap zone changes relative to the overlap zone. The variation in D-period in Figure 3 shows that levels of collagen pre-strain vary across the tissue; changes in D-period in the unstressed state have been interpreted previously as changes in internal pre-strain in articular cartilage, due to alterations in swelling pressure [17] or osmotic pressure in the tendons [45]. The D-period variations may also arise from intrafibrillar molecular tilt [52].

It is interesting to compare the SAXS data from keloid fibrotic tissue here with recent SAXS measurements on fibrotic tissue from scar tissue, skin wounds and burn tissue reported by Jiang, Tian and coworkers [27,28]. Using dehydrated tissue specimens, these researchers showed differences in the fibrillar alignment of fibrotic burn tissue with normal skin [27]. In comparing the D-period across these states [28], no significant difference was found within hypertrophic scar tissue or in comparison to normal skin, while significant differences were found for skin wounds (early and late) to normal skin, and the authors reported not being able to calculate a D-period for burns due to destruction of the nanostructure showing few characteristic D-period peaks (Figure 4 in [28]). In comparing these findings to our results, we note that the tissues in [27,28] are reported as freeze-dried, which will significantly change the peak intensity ratios with the odd orders suppressed and the even orders amplified (seen clearly for the fifth versus the sixth orders in Figure 2 in [28], for example). In contrast, our measurements on hydrated tissue exhibit strong odd-order peaks, as expected. Further, hydration influences D-period [49], so it is expected that the D-periods measured in [28] will differ (Table 1). However, it is noteworthy that considerable variability exists across the different groups of D-periods reported in [28], with the hypertrophic scar/control group in the range of 63.6 to 64.2 nm, whilst the wound tissue/control groups are in the range of 67.4 to 69.7 nm. Indeed, even the control tissue in [28] shows this variation, from 64.2 nm in scar tissue control to 68.6 nm in the wounded group control. Further, the standard deviation in D-period reported from averaged measurements in [28] are very large (up to ~8 nm), compared to the ~0.5 nm seen in the histograms in Figure 4c for our data. A possible reason for the difference could be differences in data-reduction protocols: while we fit the full peak profile to an analytic peak function with background intensity, the D-period in [28] was calculated by taking the maximum intensity and inverting it to calculate the D-period, which is susceptible to greater variability due to discretized binning in the radial profile.

Under loading, we expect that the increase in collagen peak intensity I[c] (Figure 4) is due to multiple mechanisms. Firstly, tensile strain will originally reorient fibrils away from the vertical towards the loading direction, so out-of-plane fibrils will start contributing to I[c] by intersecting the Ewald sphere. Second, stress-induced molecular or fibrillar ordering will increase the peak intensity, possibly starting from initially relaxed, partly inter-coiled “ropes” [48] of microfibrils. Lastly, stretching the tissue will reduce the scattering path in the sample due to transverse Poisson contraction, reducing I[c] (and I[ifc]). The increase in the peak intensities (Figure 4a,b), despite the transverse contraction, shows that the reorientation and ordering mechanisms are dominant. The load-induced changes in the statistical distributions for D-period (Figure 4c) show similar increasing trends across samples tested (Appendix A), but as the tissue is inhomogeneous, the D-period does not increase uniformly for all regions.

The diffraction correlation method to determine the local fibril strain and reorientation maps in the tissue with DIC was successfully implemented, with example deformation along an initial straight line described by the displaced dashed line in Figure 5a,b. The adaptation of DIC to scanning SAXS/WAXD experiments is, to the best of our knowledge, new, and possesses the important ability to bridge the heterogeneous and graded material properties of biological tissues measured with such techniques. In fact, the point-to-point correlation of diffraction and scattering patterns before and after deformation in biological systems need spatial correlation as reported herein. Indeed, it is clear from Figure 5a,b that assuming a simple, uniform deformation will give incorrect results, with physically distinct tissue voxels matched in that case. The distinct nanomechanical response at each scan point is further compounded by the spatial heterogeneity of the nanomechanical markers (like D-period) in the tissue resting state (Figure 3), as the reference state fibril pre-strain varies across the tissue as well. Such gradients in fibril D-period have recently been observed in other connective tissues such as cartilage [17]. We therefore foresee the application of the diffraction contrast method, to improve the understanding of mechanobiology and pathology in a range of graded biological structures (e.g., in the bone–cartilage interface in joints during osteoarthritic progression).

A macroscopic mechanical stress–strain curve of a keloid is shown in Appendix B
Figure A2, showing a linear behavior up to ~20% of nominal strain with a modulus (680 kPa) intermediate between the tendon and the skin (the reference lines for the tendon [53,54], dermis and hypodermis [55] layers of skin are shown on the figure). Recent atomic-force microscopy measurements report that a keloid has slightly lower values for modulus versus normal skin (~35 MPa vs. 45 MPa), but these are ~2 orders of magnitude larger than the macroscopic tissue modulus reported here [7]; small scale probes such as atomic force microscopy probes could report higher values of stiffness if weaker interfaces at larger scales are not considered.

At the nanoscale, the small fibril strain (~0.1–0.2%) relative to tissue strain may be due to interfibrillar shearing or intrafibrillar molecular tilt. As reported for tendon and mineralized collagen previously, via staggered models of fibril mechanics [56,57,58], interfibrillar shearing in a ductile matrix result in fibril strains being lower than tissue strains. Individual collagen fibrils have reported moduli of ~1 GPa (experimental ~0.4–0.5 GPa [41]; atomistic modelling 0.3–1.2 GPa [59]). The ε_F_/ε_T_ for keloids (8.5 × 10^−3^) is much less than in mineralized collagen/bone (~0.41 [37]) and tendon (~0.1–0.3 [43,60,61]), but similar to echinoderm tissue (~6 × 10^−3^ [62]). This is expected for hydrated soft tissues compared to the partially mineralized, fractal-like interfibrillar matrix in bone [63,64] or increased crosslinking [65]. The origins of low ε_F_/ε_T_ can also come from the more random fiber orientation distributions in keloids compared to tendons, with only the projected stress component onto the axial direction of fibrils. Keloid maximum collagen fibril strain (~0.17%) is lower than for bone (0.4% [37,65]), tendon (1.2 to 2.0% [43,60,66]), and cartilage (0.5–0.6% [16,17]), but comparable for echinoderms (0.2% [62]).

The other explanation for reduced change in D-period may arise from the intrafibrillar tilt, where experiments on the cornea [40] indicate that, in the early stages of tensile loading, the molecular-level reorientation of triple helical molecules in the loading direction is significant, contributing to a reorientation of the fibrils with the loading axis. Recently, a liquid crystalline elastomer model of tilted collagen fibrils has been proposed, where the percentage of D-period change is shown to be lower than that of the fibril strain when the initial tilt is substantial (i.e., for tissues such as cornea and skin, but less for tendons) [52]. Both fibrillar reorientation and interfibrillar shearing may be occurring during the tensile deformation of keloids; to clarify this, more detailed X-ray diffraction and scattering analyses may be necessary, along the lines of the model proposed in [40].

Considering the limitations, as here we are focused on method development for in situ SAXS on microscopically heterogeneous tissues, we report on a limited sample size for both scanning and in situ mechanical measurements. Nevertheless, to relate the nanomechanical parameters to tissue-level growth dynamics or in vivo stress levels, we can use the technique to analyze differences in the nanoscale structure/mechanics of keloids from different anatomically stressed tissue locations, to provide insight into how local biomechanics influence keloid pathology. Regarding the analysis, for the deformation, we focused on the meridional collagen reflections due to space limitations; adding the IFC-peak shifts would provide insight into the multi-phase, multi-directional mechanics of keloids. In addition, a 3D structural model of the fibrils and IFC components (including disorder, gap/overlap and tilting effects) would be needed to fully capture the nanoscale changes. Finally, the use of DIC on SAXS diffraction intensity maps employed fairly large subset sizes of 25 × 25 and with an overlap of 88% (step size 3 pixels) to improve continuity by sharing gray scale patterns between neighboring subsets. This was needed due to the availability of quite large tissue deformation steps (0 to 20%) for the correlation, thus allowed only a preliminary mapping of full-field strain. The mapping was, in any case, proven sufficiently accurate to interpret and link SAXS reading as previously shown. Future work will refine the digital image correlation/SAXS method by including more deformation steps, and integrating the method with dual imaging and diffraction (DIAD) data.

## 5. Conclusions

In conclusion, in this proof-of-concept study, we have shown how microbeam synchrotron SAXS, in situ tensile testing, and image correlation can be combined to measure fibrillar-level strain and reorientation in microscopically inhomogeneous, textured, collagenous tissues, using keloid scar tissue as an example. Our approach can be applied to understand the role of nanoscale biophysical forces in fibrotic progression in multiple disorders (e.g., dysfunctional wound healing, burn injuries, or scleroderma), allowing the elucidation of mechanobiological pathways and structural biomarkers.

## Figures and Tables

**Figure 1 materials-15-01836-f001:**
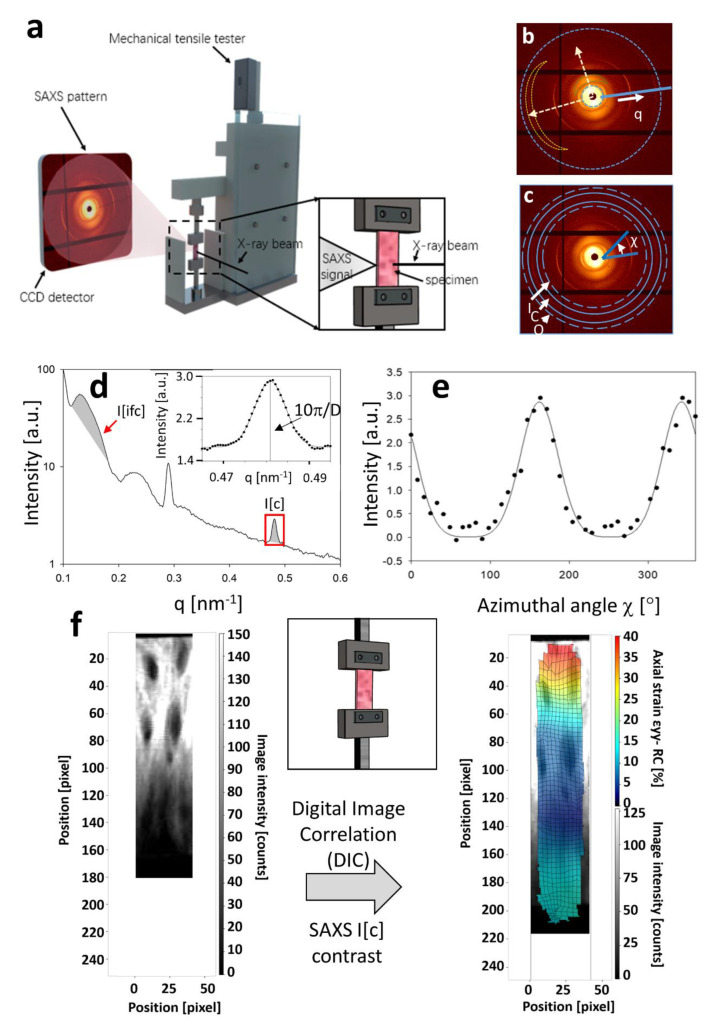
In situ mechanics with small angle X-ray scattering (SAXS) and experimental data analysis: (**a**) *experimental setup*: Mechanical tester in X-ray beam path (transmission geometry), with a representative SAXS image on the CCD detector. Right inset: enlarged view of sample holder, specimen and with X-ray beam. Tensile strain by vertical displacement of upper sample holder; (**b**,**c**) *Keloid SAXS pattern with integration modes* (**b**) Radial I(q) and (**c**) azimuthal I(χ) profiles. In (**b**) crescent dashed shapes show the 2nd IFC peak (green) and 5th collagen peak (yellow). In (**c**), *I*, *C*, and *O* denote the inner, centre and outer rings for I(χ) calculation; (**d**) *Example I(q) profile* showing 2nd order IFC (grey shading, left) and 5th order collagen (grey shading, right); inset: Gaussian−exponential fit to collagen peak; (**e**) *Example I(**χ**) profile* with Gaussian model fit data; (**f**) Conceptual schematic of the diffraction contrast image correlation method for SAXS mapping. *Left*—undeformed example SAXS intensity map from collagen peak (after background correction), showing spatial contrast in nanoscale texture; *Right*—deformed grid of tissue displacements based on correlating collagen peak SAXS intensity before and after; note DIC grid resolution lower than SAXS microfocus step.

**Figure 2 materials-15-01836-f002:**
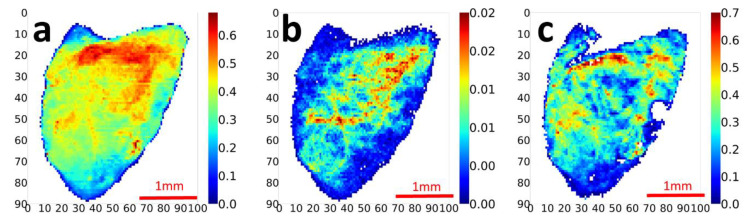
Contrast 2D mapping images with SAXS show complementary collagenous and interfibrillar peak concentrations: Color maps with each SAXS pattern are obtained by calculating different peaks in the radial intensity profile plot for (**a**) whole intensity mapping image, (**b**) Collagen I[c] background-corrected intensity and (**c**) IFC I[ifc] background-corrected intensity. Here and in following figures, the x- and y-axes are in scan number indices.

**Figure 3 materials-15-01836-f003:**
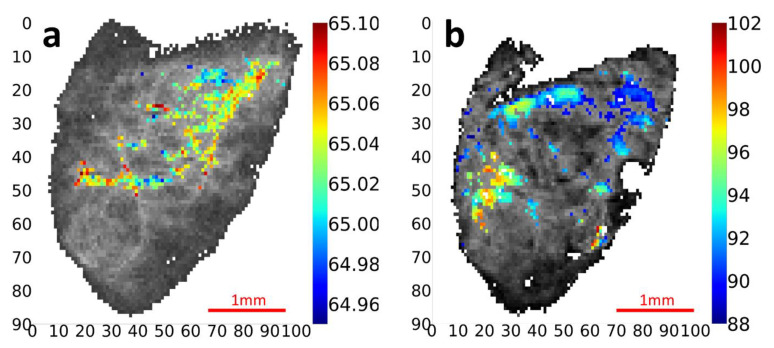
SAXS Contrast 2D mapping of collagen D-period and interfibrillar spacing peaks: For the points with high intensity of collagen or proteoglycan, pre-strain of each point has been calculated. Underlying gray scale intensity mapping is shown as reference (**a**) collagen D-period mapping and (**b**) proteoglycan pre-strain mapping. Scale bar units are in nanometers (nm).

**Figure 4 materials-15-01836-f004:**
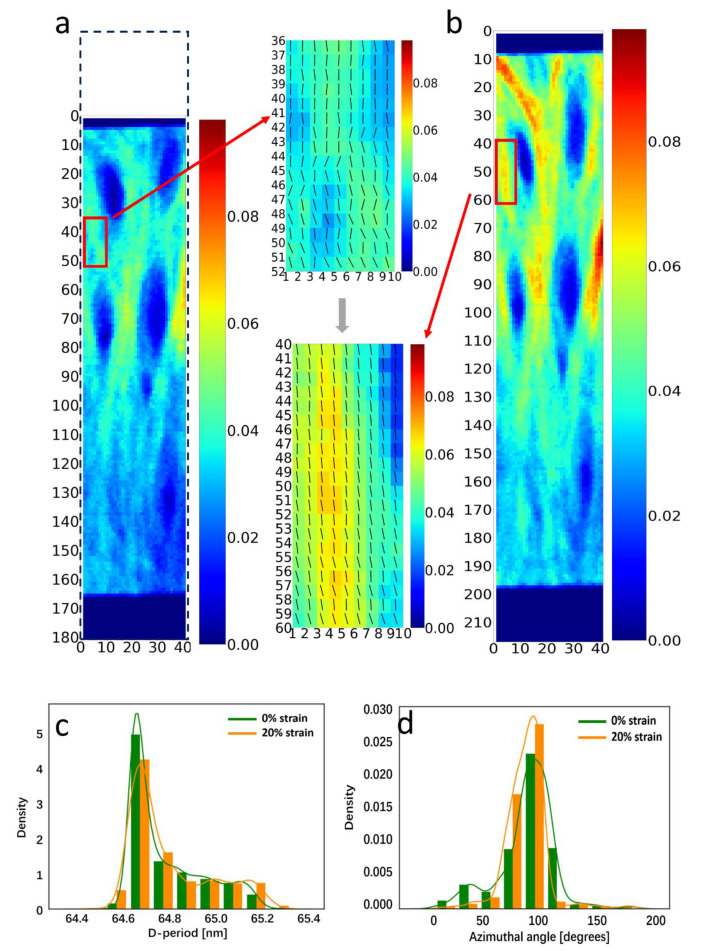
In situ tensile strain leads to variation of SAXS intensity contrast: (**a**) at 0% strain, collagen intensity mapping with the enlarged region of interest; dashed line is the scan size at 20% strain; (**b**) at 20% strain on same sample, collagen intensity mapping with the enlarged region of interest; (**c**) histogram of collagen D-period distribution at 0% and 20% strain; (**d**) histogram of collagen orientation angle at 0% and 20% strain. Smooth lines: kernel density estimators.

**Figure 5 materials-15-01836-f005:**
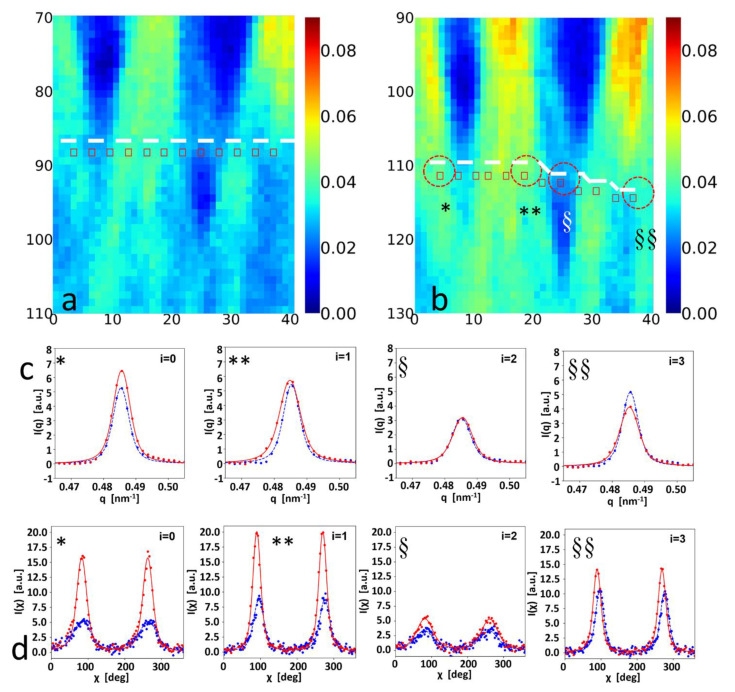
Digital image correlation (DIC) of SAXS contrast mapping for local fibrillar–level mechanics: (**a**,**b**) Colour–maps of background-corrected collagen peak intensity (used for DIC mapping) from keloid cross-sections in reference (left (**a**): 0%) and strained states (right (**b**): 20%). Left (**a**), red rectangles: selected set of SAXS patterns, on a horizontal line (dashed). DIC displacements (*u*_x_, *v*_y_) for each selected point used to calculate the displaced positions (right (**b**), red rectangles) and the distorted line (dashed). *, **, §, §§: selected points for detailed comparison in (**c**,**d**). (**c**): I(q) intensity profile around the 5th order collagen peak at 0% (blue: colour online) and 20% (red: colour online) strain for points at *, **, § and §§. *: increase in peak intensity, no peak shift; **: shift of peak to lower q, peak broadening; §: minimal change; §§: decrease in peak intensity. (**d**): Corresponding I(χ) changes: *: decrease in peak width; **: increase in peak width; §: minimal change; §§: shift of peak aligning to the vertical direction.

**Figure 6 materials-15-01836-f006:**
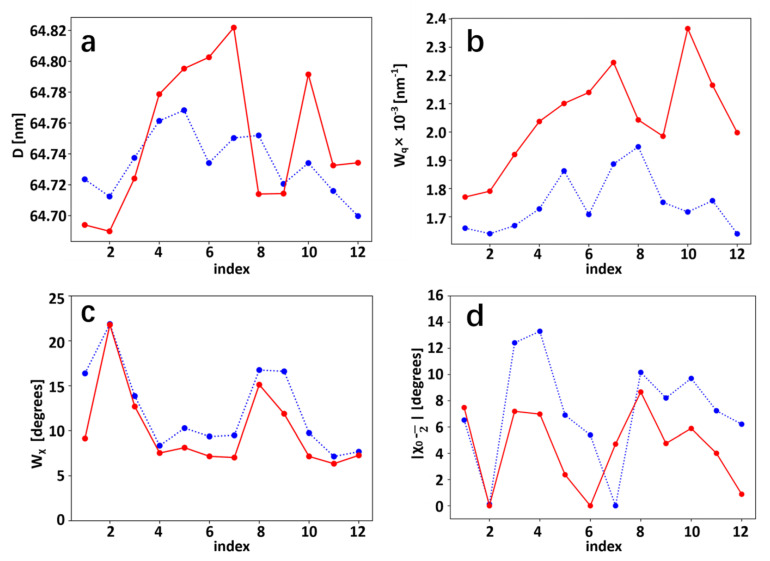
Fibrillar D-period, axial and azimuthal peak widths, and angular position before and after loading: For the selected line-scan (red rectangles) in Figure 5a, line plots of D, w_q_, w_χ_ and deviation of mean fibril angle χ_0_ from π/2 (for all points) are shown before; (**a**) D-period predominantly increases on tension (as expected), but substantial intra-slice variability is seen. (**b**) variability in D-period (w_q_), showing a consistent increase; (**c**) consistent but limited decrease in width of fibril angular distribution w_χ_ on tension, and (**d**) nearly consistent alignment of fibrils towards the loading axis. *, **, § and §§ denote same subset of points whose intensity profiles are shown in Figure 5.

**Table 1 materials-15-01836-t001:** Top: meridional peak intensity ratios (odd/odd: (I3/I5), and even/odd: (I4/I5) and (I6/I5)) for several connective tissues taken from the literature, and keloid tissue from the current work. Bottom: comparison of native D-period of the same set of tissues (with bone added) to keloid tissue.

Tissue	I_3_/I_5_	I_4_/I_5_	I_6_/I_5_	Reference
cartilage	2.99	0.15	0.09	[16]
wet tendon	3.02	0.39	0.46	[43]
skin	4.37	0.69	0.67	[25]
scar tissue	-	-	~7 *	[28]
keloid	6.70	0.05	0.07	current work
dry tendon	2.24	2.35	9.18	authors’ data
**Tissue**	**D-Period**		**Reference**	
cartilage	66 nm		[16]	
wet tendon	67 nm		[21,44]	
skin	64.6 nm		[25]	
scar tissue	63.3–64.4 nm		[28]	
skin wound	67.4–69.7 nm		[28]	
keloid	64.7—65.3 nm		current work	
dry tendon	65 nm		authors’ data	
bone	65.5–66.4 nm		[19,24]	

not measurable from data in reference; *: approximate read.

## Data Availability

Data used to generate the SAXS intensity and other color-maps, as well as the I(q) and I(χ) curves shown, are available on request.

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
