# Peer review of "Investigating the Fibrillar Ultrastructure and Mechanics in Keloid Scars Using In Situ Synchrotron X-ray Nanomechanical Imaging"

_materials, 2022, doi:10.3390/ma15051836_

Round 1

Reviewer 1 Report

Zhang et al., submitted the paper entitled “The fibrillar ultrastructure and mechanics in keloid scars using in situ synchrotron X-ray nanomechanical imaging” to publish in “Materials (I.F= 3.623)”. In which, by using SAXS and DIC studies, the author described the fibrillar structural property and mechanics involved in the keloid scars. It is a nice paper, it can be accepted after addressing the queries.

  1. Introduction: Its looks too elaborate, which may affect the readership. Author must concise the introduction part with essential information and current challenges and the problem that is focused/resolved in this manuscript.
  2. Many abbreviations are mentioned repetitively throughout the manuscript. For example, extracellular matrix (ECM) has been mentioned in many sentences. Author must avoid those repeating abbreviations, better to mention as ECM.
  3. Scale bars in many Figures are not at readable fashion, thus resolution must be improved in the revised version.
  4. Appendix A is not clarified in the text.

Reviewer 2 Report

Title: The fibrillar ultrastructure and mechanics in keloid scars using in situ synchrotron X-ray

nanomechanical imaging

1. The study presented in the manuscript tries to quantify (at the microscale) fibrillar reorientations, which authors suggest to increases in fibrillar D-period variance, and increases in mean D-period under macroscopic tissue 31 strains of ~20%. Although, it is a well-written manuscript with few too-complex sentences, which readers would find difficult to understand. The major limitation of the manuscript is the sample size (n=2). There is no statistical analysis conducted to appreciate the reproducibility of the study.

The research claims that it is proof-of-concept where they have used microbeam synchrotron SAXS, in situ tensile testing and image correlation to uncover the collagenous ECM structure-function relations in keloid scar tissue at the ultrastructural level. But the results do not suggest that, as only two samples were used and no statistical analysis was conducted.

2. It would be relevant and exciting if it was supported by statistically significant data size.

3. The microbeam synchrotron SAXS has been reported before, the only originality is that it is combined with tensile testing.

4. This subject area is relevant and could have added value if it was presented with some rigorous statistical tests.

5. the paper is well written, but few places too complex sentences that may be difficult for readers.

6. I disagree with the conclusions the evidence and arguments presented; it is very hard to conclude by using two samples. Authors’ mentions in conclusion section that images presented in the manuscript uncover the collagenous structure-function at ultrastructural level, but how? It is not very clearly presented in the manuscript. 

7. they do not really address the main question posed.
